# Biocompatibility Evaluation and Enhancement of Elastomeric Coatings Made Using Table-Top Optical 3D Printer

**Giedre Grigaleviciute** [1], **Daiva Baltriukiene** [2], **Virginija Bukelskiene** [2] and **Mangirdas Malinauskas** [1,*]

[1]  Laser Research Center, Faculty of Physics, Vilnius University, Vilnius LT-10223, Lithuania; giedre.grigaleviciute@gmail.com

[2]  Life Sciences Center, Institute of Biochemistry, Vilnius University, Vilnius LT-10257, Lithuania; daiva.baltriukiene@bchi.vu.lt (D.B.); virginija.bukelskiene@bchi.vu.lt (V.B.)

*  Correspondence: mangirdas.malinauskas@ff.vu.lt; Tel.: +370-60002843

**Abstract:** In this experimental report, the biocompatibility of elastomeric scaffold structures made via stereolithography employing table-top 3D printer Ember (*Autodesk*) and commercial resin FormLabs Flexible (*FormLabs*) was studied. The samples were manufactured using the standard printing and development protocol, which is known to inherit cytotoxicity due to remaining non-polymerized monomers, despite the polymerized material being fully biocompatible. Additional steps were taken to remedy this problem: the fabricated structures were soaked in isopropanol and methanol under different conditions (temperature and duration) to leach out the non-polymerized monomers. In addition, disc-shaped 3D-printed structures were UV exposed to assure maximum polymerization degree of the material. Post-processed structures were seeded with myogenic stem cells and the number of live cells was evaluated as an indicator for the material biocompatibility. The straightforward post-processing protocol enhanced the biocompatibility of the surfaces by seven times after seven days soaking in isopropanol and methanol and was comparable to control (glass and polystyrene) samples. This proposes the approach as a novel and simple method to be widely applicable for dramatic cytotoxicity reduction of optically 3D printed micro/nano-scaffolds for a wide range of biomedical studies and applications.

**Keywords:** stereolithography; elastomer; biocompatibility; post-processing; UV curing; thermal treatment; optical 3D printing

## 1. Introduction

Recently, diverse 3D printing technologies have received a lot of attention in the areas of science and industry [1]. Differently from traditional processes of manufacturing, 3D printing allows computer aided manufacturing (CAM) of arbitrary geometry objects using computer aided design (CAD) models out of variety of materials with minimal fabrication costs. It can be applied in various fields such as mass-customized production [2], prototyping, and dentistry [3]. One of the most promising is regenerative medicine, where 3D printed objects have already made a great influence in such areas as orthopedics [4]; face and skull reconstruction [5]; plastic, teeth, and mouth surgeries [6]; etc. Another 3D printing application area in medicine is tissue engineering [7]. For this purpose, cells can be seeded into well-defined geometry 3D printed structures [8–10] and artificial tissue or organs can be grown and implanted into a living organism [8,11].

A broad range of biomaterials depending on their chemical structure and mechanical and biological properties are used in 3D printing [12,13]. However, there is still a lack of elastomeric

photostructurable resins that fulfill the tissue engineering requirements for scaffolds [7,14]. This significantly influences a demand to synthesize novel biopolymers with tunable bio-properties (mechanical, wetting, bioresorptive, etc). On the other hand, the optimizing procedure of the biocompatibility of existing commercial resins could also significantly contribute to the wide-spread use of the technique in advanced clinics [15]. However, the composition of commercial resins is proprietary. It is complicated to predict how these polymers will affect cells and their surrounding microenvironment. Consequently, there is a demand for comprehensive polymer processing and biological testing protocols [16].

It was shown that cell viability can be increased by higher polymerization degree or monomer-to-polymer conversion level [17]. For this reason, additional UV exposure after polymerization could be a solution in order to optimize biocompatibility. Routinely, polymerized structures are soaked in water or other polar solvents such as isopropyl alcohol, methanol, and ethanol for up to 24 h to allow uncured monomers to be leach out. Usually, this procedure is insufficient to ensure high biocompatibility of fabricated structures [18]. It was shown that prolonged soaking of polymerized structures might result in better cell viability because of leaching out of monomers that are toxic to cells [19]. In other attempts to mitigate the cytotoxicity of polymers, treatment with supercritical carbon dioxide drying [20], washing in 99% ethanol, and coating with wax [21], biocompatible hydrogels, or PDMS [22] were used. However, in these reports, the optimal biocompatibility protocol was not defined as the research was not systematic. Therefore, new research efforts aiming at the development of effective post-manufacturing procedures are necessary as there exist very few and non-comparable published data.

Here, we report a straightforward procedure for optimizing the biocompatibility of 3D printed structures using a commercially available optical 3D printer and common resin. The protocol includes simple additional soaking in methanol and isopropanol under different conditions and then extra UV exposure. The achieved improvement in biocompatibility significantly reaches up to seven times and is comparable to glass/polystyrene, which are known as biocompatible reference materials.

## 2. Materials and Methods

During the experiments, a 3D printer *Autodesk Ember*, which provides projection stereolithography based on a digital mirror device shaping the light source of 405 nm wavelength and 5 W power layer-by-layer, was used. Disk-structures were fabricated out of commercial *Formlabs Flexible* photoresin, which consists of acrylate monomers and oligomers and owns elastomeric properties [23]. The density of *Formlabs Flexible* photoresin is 1.09–1.12 g/cm$^3$, boiling point >100 °C, viscosity 4500 cps at 25 °C temperature. Some of mechanical properties of this material are shown in Table 1 as provided by the producer [23].

**Table 1.** Some of mechanical properties of *Formlabs Flexible* material [23].

| Mechanical Property | Formlabs Flexible | |
| --- | --- | --- |
| | Resin | Cured Object |
| Tensile strength | 3.3–3.4 MPa | 7.7–8.5 MPa |
| Elongation | 60% | 75–85% |
| Tear force | 9.5–9.6 kN/m | 13.3–14.1 kN/m |

Several default printing parameters were changed for the *Formlabs Flexible* processing: layer height of 0.025 mm, wait (before exposure) of 6 s, exposure time of 14 s, and separation velocities of 1 rpm. The disk-shaped structures of 13 mm diameter were polymerized and post-processed to test biocompatibility (Figure 1). Additional UV exposition (using Thorlabs CS2010 UV LED lamp, wavelength 365 nm, and power 270 mW) of two different durations of 1 and 22 h was implemented to increase polymerization degree [24]. In addition, leaching out the not polymerized monomers that are toxic for cells [19] was carried out when soaking fabricated structures in isopropyl alcohol and

methanol in two different temperatures (22–25 °C and 37–40 °C) and different durations of from one to eight days. Experiment plan is shown in Figure 2.

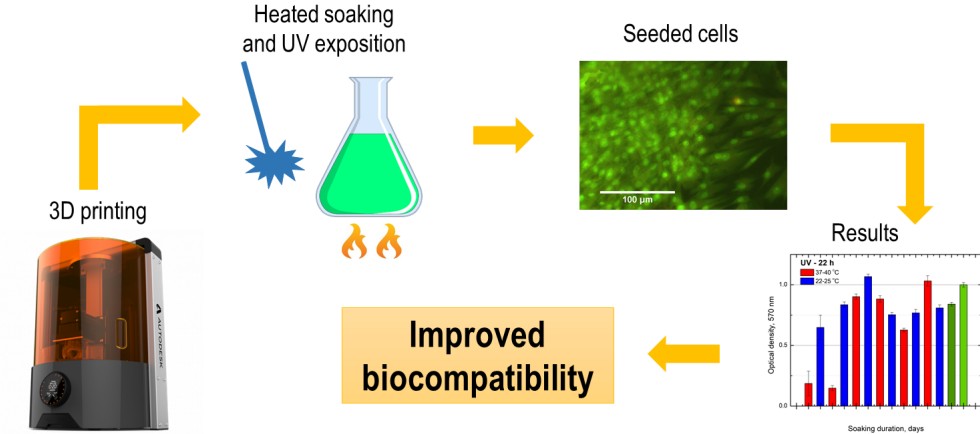

**Figure 1.** Steps of the experiment: first, samples were 3D printed; secondly, additional UV exposure and heated soaking was implemented; third, samples were seeded with cells and biocompatibility tests were made; and, fourth, results were discussed and conclusions made.

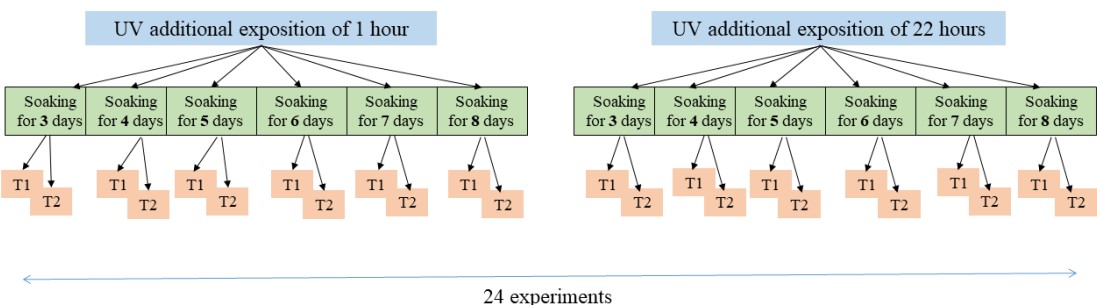

**Figure 2.** Experiments post-processing plan scheme: T1, 22–25 °C ; T2, 37–40 °C. First, polymerized structures were additionally UV exposed for 1 or 22 h. Then, samples were soaked in two different solvents for different duration from one to eight days at two different temperatures.

The viability of cells was determined using the standard MTT [3-(4,5-dimethylthiazol-2-yl)-2,5-diphenyltetrazolium bromide] assay (Sigma, St. Louis, MO, USA). Myogenic stem cells were seeded on samples at a density of 30,000 cells/mL/sample, using glass slides as control. The samples were placed in 24-well polystyrene tissue culture plates and incubated at 37 °C with 5% $CO_2$ atmosphere. After 24 h of culture, the medium was discarded and the samples were treated with MTT (1 mg/mL) and incubated for 1 h at 37 °C. The MTT solution was then carefully replaced with 200 μL of DMSO to solubilize the formazan and, subsequently, 100 μL of the formazan-DMSO solution was used to measure absorption. The optical density at 545 nm was measured by using an automatic microplate reader Varioskan Flash (Thermo Scientific, Vantaa, Finland). Results were calculated as the ratio of cells grown on tested materials to polystyrene tissue culture plate surface.

To assess the mode of cell death, myogenic stem cells were seeded on samples at a density of 50,000 cells/mL/sample. The samples were then incubated at 37 °C with 5% $CO_2$ atmosphere. After being cultured for 24 h, cells were trypsinized (0.25% trypsin). Then, 25 μL of cell suspensions were transferred to glass slides. Dual fluorescent staining solution (2 μL) containing 100 μg/mL acridine orange and 100 μg/mL ethidium bromide (AO/EB, Sigma, St. Louis, MO, USA) was added to each suspension and then covered with a coverslip. The morphology of apoptotic cells was examined and

100 cells were counted using a fluorescent microscope (Olympus, Tokyo, Japan). Acridine orange was used to characterize chromatin condensation and segmentation; ethidium bromide was used to characterize membrane integrity, as described in [25]. Cells were categorized as follows: viable, viable apoptotic, nonviable apoptotic, and necrotic.

Statistical analysis was performed using R Studio by one-way analysis of variance (One-way ANOVA) with post-hoc Tukey HSD. Viability graphs were made with GraphPad Prism software. Data are presented as means and verified by at least three independent experiments. A value of $p < 0.05$ was considered to be statistically significant. The significant differences are stressed with symbols in figures.

## 3. Results

In this study, muscle-derived stem/progenitor cells were used for biocompatibility assessment. These cells are multipotent cells, demonstrating high self-renewal and long-term proliferation capacities. These cells are able to promote endogenous tissue repair by secreting trophic factors [26–28]. Therefore, they are a valuable source for regenerative medicine and tissue engineering applications. In cell culture, myogenic stem cells assume spindle-shaped fibroblastic appearance, rapidly proliferate, and become confluent in less than one week [29]. Recently, cell shape has emerged as determinant, which controls cell proliferation, growth, physiology, and adaption for specific functions [30]. The cells also change their morphology in response to toxic stimuli. To assess whether 3D printed samples can affect cell shape, we examined morphological changes of cells grown on differently processed samples. Scanning electron microscopy (SEM) analysis confirmed (Figure 3) a typical spindle-shaped cell phenotype. Moreover, almost confluent cover of the surfaces tested was detected after 24 h post-seeding.

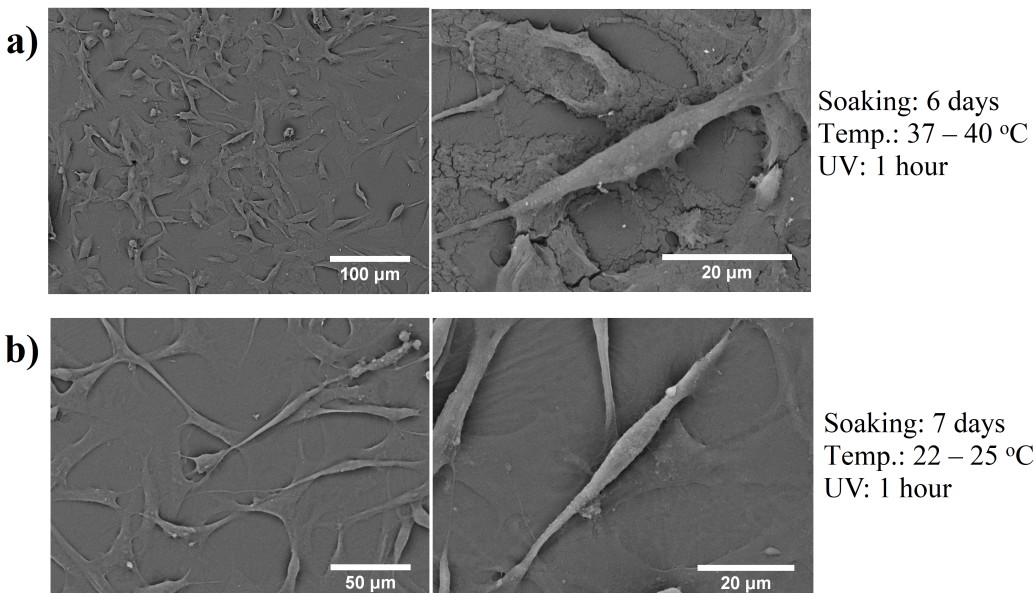

**Figure 3.** SEM analysis (representative images) of cells grown on different samples: (**a**) soaking duration in heated isopropanol and methanol for six days; and (**b**) soaking duration in non-heated isopropanol and methanol for seven days. Note that the sample surfaces do not influence the morphology of the cells.

Next, cell viability was analyzed to assess overall cell response to the tested samples. Data show that cell viability increased with increasing soaking time. An optimal time for samples preparation in isopropanol and methanol was a minimum of five days. At Day 7, the maximal biocompatibility was reached. The effect of UV exposure was rather negligible. UV curing for 1 h increased cell viability

60-8% and was comparable to control (Figure 4). However, the monomer cross-linking effect tended to be saturated, as the cell viability did not remarkably increase after longer (22 h) exposure time.

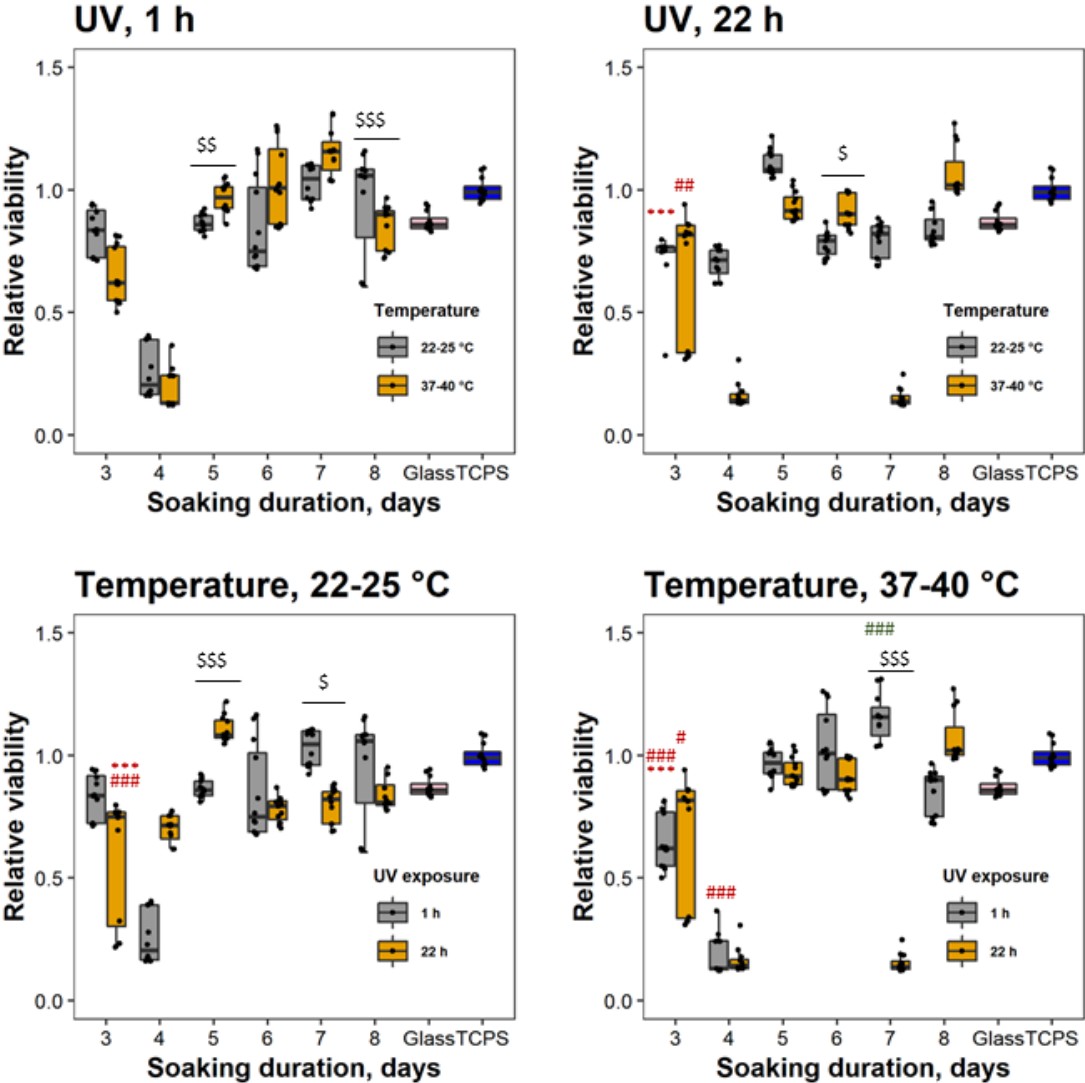

**Figure 4.** Biocompatibility test results. (**top**) The relative viability depending on soaking duration and temperature, when UV exposition was 1 and 22 h. (**bottom**) The relative viability depending on soaking duration and UV exposition, when temperature was 22–25 °C and 37–40 °C. *, # mark statistically significant changes then *p* < 0.05, ** *p* < 0.01, *** *p* < 0.001. * and # mark significant differences between tissue culture plate surface (TCPS) and glass surface, respectively. $ marks significant differences between samples processed under the same time point. Red signifies differences below 1 (control level, TCPS) and green differences above 1.

The samples had a modest effect on overall cell viability in a post-processing dependent way. Therefore, the mode of cell death was evaluated subsequently (Figure 5). The higher number of dead cells was detected in the population of cell grown on samples prepared at 37–40 °C. We hypothesize that this phenomena can be determined by the material biodegradation under physiological temperature. However, our results show that this negative effect on cell viability may be reduced by the additional UV exposure with subsequent longer soaking in polar solvents. In all cases, necrosis was a dominant mode of cell death.

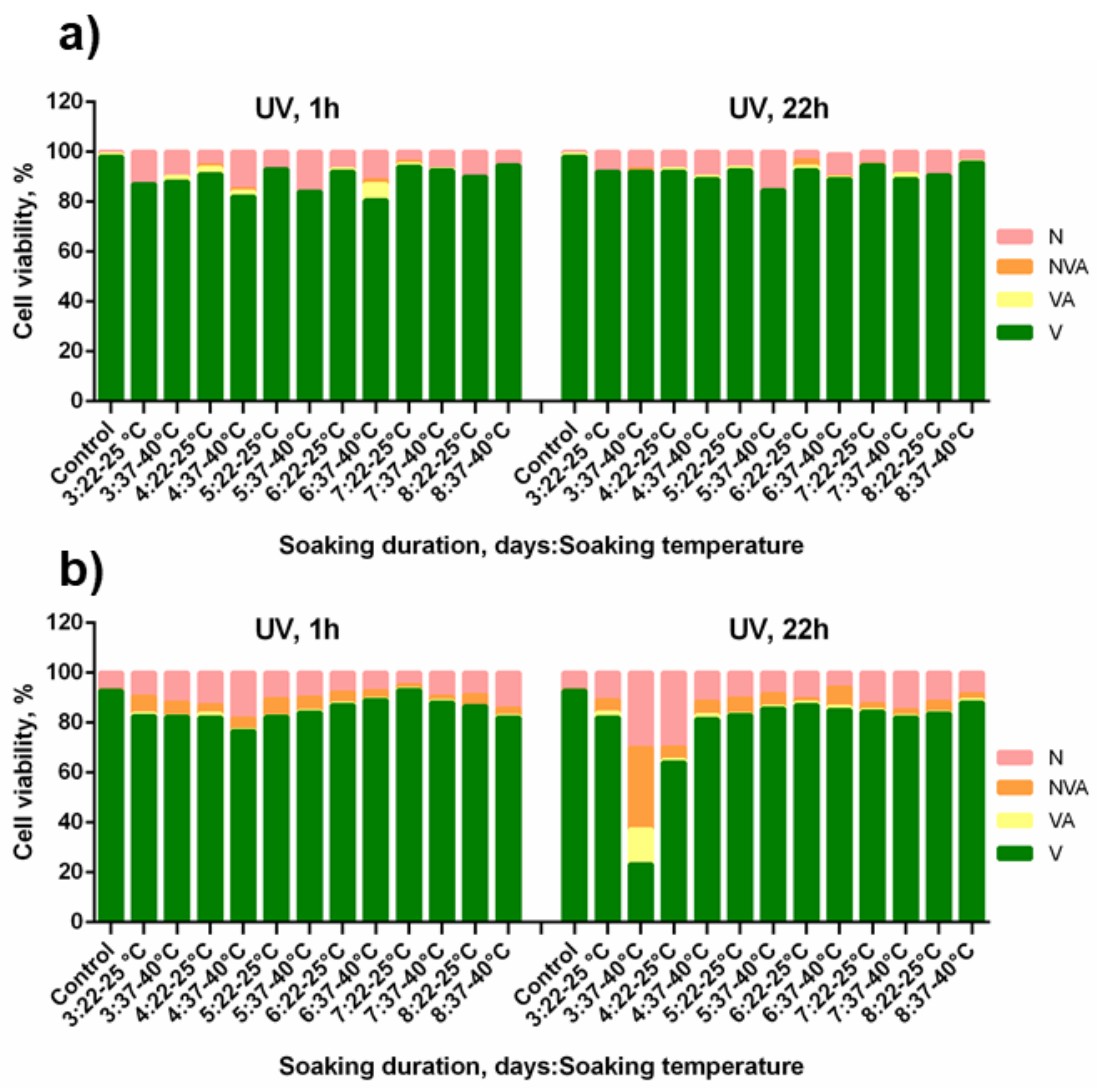

**Figure 5.** Cell viability results. The percentages of cells were measured after 24 h (**a**) and 48 h (**b**) post-seeding. V, viable cells; VA, early apoptosis; NVA, late apoptosis; N, necrosis.

## 4. Discussion

Needless to say that modern trend of interdisciplinarity can lead to these results being relevant to not only 3D-bioprinting, but also as a supplement to other 3D printing techniques. Indeed, various combinations of additive–additive [9,31] and additive–subtractive [32,33] manufacturing techniques were used in the past to great effect. For instance, laser induced forward transfer (LIFT) [34] could be used to directly and selectively seed scaffolds with cells [9]. The 3D femtosecond laser nanolithography could make sub-micrometer additions to the macro-structure of produced scaffolds [31], out of huge variety of different materials, including non-photosensitized [35,36] or fuctionalized [37] polymers. In essence, pairing of various processing techniques is a powerful way to offset most of the technological drawbacks and accentuate advantages [1,33]. For instance, 3D laser nanolithography is struggling to process elastic materials [14,38]. Thus, by pairing it to stereolithography would allow making macro-level elastic structures that would require only minor laser made additions. The majority of currently used synthetic elastomers are reproducible, cost effective, and excellent alternatives to extracellular matrix. However, they have to meet high biocompatibility (i.e., must elicit a negligible immune reaction and cells must adhere, migrate onto the surface and through the scaffold, proliferate, synthesize new matrix, and begin to differentiate) as it is critical for any biological and biomedical

application [39]. Cell and tissues can be affected by toxic compounds released from polymer due to its degradation or extraction by biological fluids [16,40]. To enhance the biocompatibility of the materials used for the fabrication of tissue engineering scaffolds and other biomedical applications, surface modification strategies, such surface grafting, abrasive blasting, acid etching, surface coatings, and heat and steam treatment are employed by several research groups [41]. These modifications alter surface roughness, hydrophilicity, surface charge, and have direct effect on biocompatibility and cell fate. On the other hand, there is a lack of available information on a composition of commercial resins used for 3D printing (mostly intentionally due to high market competition). Therefore, it is difficult to predict how specific polymers will affect cell fate and which effect it will have on human health and environment. Toxic compounds are released from polymers in relatively small amounts. Therefore, short-term processing of biopolymers are less relevant [16]. Here, we report more cost effective way to enhance elastomer biocompatibility which includes prolonged soaking in polar solvents and additional UV exposure. Data suggest that proper exploitation of this procedure may quench the thirst of long-time unmet demands for biocompatibility.

The combination of additional UV exposure and long-duration soaking might be helpful for structured surfaces too. We have tested additional UV exposure and soaking on chemically synthesized polymers including porous structures and obtained promising data. Therefore, if the UV exposure did not reach the inner layers of the structure, a prolonged soaking in the polar organic solvents such as methanol and isopropyl alcohol would reach the non-polymerized monomers and leach them out.

## 5. Conclusions

A novel and straightforward protocol for improving the biocompatibility of 3D printed polymer objects (or coatings) was introduced and experimentally validated. The specific biocompatibility of commercial optically 3D printed elastomeric resin *Formlabs Flexible* surfaces using table-top *Ember* device was improved by implementation of additional UV exposure, heating, and prolonged soaking in isopropanol and methanol. No samples had a remarkable effect on cellular morphology, yet the majority of the samples' cell viability was higher than 85% (comparable to polystyrene substrates). Moreover, a longer soaking duration, comparing four days (the sample with the worst biocompatibility) and seven days (the sample with the best biocompatibility) tends to reveal more than seven times higher biocompatibility in the test results. In brief, the table-top 3D printer fabricated and post-processed samples can be of high biocompatibility applying the proposed method due to higher polymerization degree and the leaching out of the non-polymerized monomers. These findings propose the further application of such approach for all kinds of polymer objects for basic cell studies and practical tissue engineering—not limiting to surfaces or coatings, but also free-form meshes and complex-shaped 3D scaffolds.

**Author Contributions:** Conceptualization, G.G. and M.M.; methodology, G.G., D.B., and M.M.; formal analysis, all authors; investigation, all authors; resources, D.B. and M.M.; data curation, G.G., D.B., and V.B.; writing—original draft preparation, all authors; writing—review and editing, G.G., D.B., and M.M.; supervision, D.B. and M.M.; project administration, D.B.; and funding acquisition, D.B. and M.M. All authors have read and agreed to the published version of the manuscript.

**Funding:** This research was funded by the Research Council of Lithuania (Grant No. SEN-13/2015).

**Acknowledgments:** G.G. sincerely acknowledges Edvinas Skliutas (VU LRC) for the UV exposure measurements and comparative calculations.

**Conflicts of Interest:** The authors declare no conflict of interest.

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
