# Peer review of "Biocompatibility Evaluation and Enhancement of Elastomeric Coatings Made Using Table-Top Optical 3D Printer"

_coatings, doi:10.3390/coatings10030254_

Round 1

Reviewer 1 Report

In the paper “Biocompatibility evaluation and enhancement of elastomeric coatings made using table-top optical 3D printer” by Malinauskas el al. the authors describe a protocol to improve biocompatibility that of a 3D printed layer of the commercial resin (FormLabs).

The work aims to tackle an important aspect for the biomaterial community which is the possibility to improve biocompatibility by means of non-destructive sterilization protocols. However, I think that some effort should be done to improve the discussion about the universality of the proposed procedure and the impact that this can have in material science.

My main questions are:

- Which is the reason to use myogenic stem cells to assess the biocompatibility of this material?

- Is it possible that photodamage of the resin occurs after 22h or UV exposure? What happens between 1h and 22h? I do not understand why the cell viability is not increased after a long UV exposure time.

- Which are the “tissue engineering for scaffolds” requirements the authors refer to at pag 1 line 30?

- Can the authors comment on the effect of their protocol on a more structured surface?

- Which kind of biodegradation effect is the one cited at pag 4 line 103? Have the authors performed some surface material characterization analysis to support this idea?

- What happens after 7 days of soaking time? Do you have any idea of what can cause a reduction in cell viability?

Author Response

Answers to Reviewer 1

In the paper “Biocompatibility evaluation and enhancement of elastomeric coatings made using table-top optical 3D printer” by Malinauskas el al. the authors describe a protocol to improve biocompatibility that of a 3D printed layer of the commercial resin (FormLabs).

The work aims to tackle an important aspect for the biomaterial community which is the possibility to improve biocompatibility by means of non-destructive sterilization protocols. However, I think that some effort should be done to improve the discussion about the universality of the proposed procedure and the impact that this can have in material science. My main questions are:

Comment R1.1 Question - Which is the reason to use myogenic stem cells to assess the biocompatibility of this material?

Answer to R1.1 Muscle stem cell have been identified as a valuable source for regenerative medicine and tissue engineering applications because of their ability to differentiate into other cell lineages such as osteocytes, adipocytes or even neural cells. These cells possess high myogenic capacity and effectively regenerate both skeletal and cardiac muscle. They can be applied for the treatment of various musculoskeletal, cardiovascular, and urological disorders. Currently, we are looking for relevant synthetic polymer for soft tissue engineering. Therefore, these skeletal muscle derived cells were chosen as an excellent candidate for our studies. The corresponding text was added in the Results Section.

Comment R1.2 - Is it possible that photodamage of the resin occurs after 22h or UV exposure? What happens between 1h and 22h? I do not understand why the cell viability is not increased after a long UV exposure time.

Answer to R1.2 It is known that monomers can be toxic to the cells. During the photopolymerization process there might be left a number of non-polymerized monomers in the structure. The additional UV exposure increases the degree of cross-linking and the toxic non-polymerized monomers can be eliminated this way. Theoretically, the longer UV exposure - the bigger exposure dose and the higher cross-linking degree. If all non-polymerized monomers are cross-linked, longer UV exposure gives no additional benefit for cell viability - this process tends to saturate. The UV dose used for exposure is too low for photodamaging the resin surface. Namely, the UV diode accumulated exposure during 1h was lesser than the one of 14s exposure while 3D printing using Ember device. On the other hand, the 22h exposure dose was higher by 1 order of magnitude in comparison to Ember’s, but taking into account the much lower intensity (incident light power per normalized area) this still should not be the case inducing the photodamage. From our experimental experience only pulsed laser UV irradiation (even if non-focused, but being of several orders higher intensity) generates significant photodamage of resins, then the parameters should be chosen precisely and responsibly.

Comment R1.3 - Which are the “tissue engineering for scaffolds” requirements the authors refer to at pag 1 line 30?

Answer to R1.3 Scaffolds used for tissue engineering applications should (i) be biocompatible cells must adhere, function normally, migrate onto the surface and eventually through the scaffold and begin to proliferate before laying down new matrix, and support cell differentiation; after implantation scaffolds must elicit negligible immune reaction; (ii) have relevant mechanical properties consistent with the anatomical site into which it is to be implanted, and must be strong enough to allow surgical handling during implantation; (iii) have to be easy manufacturable to create a porous (with interconnected pores) structure to ensure cellular penetration and adequate diffusion of nutrients to cells within the construct and to the extra-cellular matrix formed by these cells; and finally (iv) be cost effective and ensure a possibility to scale-up from making one at a time in a research laboratory to small batch production. According to the answer addition of text was made in the amended Discussion part.

Comment R1.4 - Can the authors comment on the effect of their protocol on a more structured surface?

Answer to R1.4 Indeed, the combination of additional UV exposure and long-duration soaking might be helpful for structured surfaces too. Additional UV exposure and soaking, we have tested on chemically synthesised polymers including porous structure and got a promising data. Therefore, if the UV exposure did not reach the inner layers of the structure, a prolonged soaking in the polar organic solvents such as methanol and isopropyl alcohol would reach the non-polymerized monomers and leach them out. A following paragraph was added into the Discussion Section.

Comment R1.5 - Which kind of biodegradation effect is the one cited at pag 4 line 103? Have the authors performed some surface material characterization analysis to support this idea?

Answer to R1.5 Currently, most biodegradable polymers are broken down by hydrolysis, resulting in the accumulation of acids that may alter the pH of the microenvironment or exert more direct toxicity. Some polymers can be destroyed by macrophages, inducing an inflammatory reaction. We used commercial resin, which detail chemical composition is unknown. However, incubation of the samples in cell culture medium did not change its pH. Therefore, we supposed that in this case reduced cell viability could be associated with leaching out of toxic monomers. Verily, our results showed that negative effect on cell viability may be reduced by additional treatment.

Comment R1.6 - What happens after 7 days of soaking time? Do you have any idea of what can cause a reduction in cell viability?

Answer to R1.6 Indeed, on day 7, we have registered a remarkably reduced cell viability, which was rather exception as on day 8 cell viability was even higher than viability of cells grown on control surfaces.It was reported that the polymer specimens remain unchanged until they reach a critical time point (Davison et al. Tissue engineering 2014, p. 177-215).In this case, the reduced cell viability had to be detected on day 8, also. Perhaps, our samples overcome this critical time point, as we repeated the experiment 3 times and got the same tendency.

Reviewer 2 Report

This paper provides a novel way to fabricate biocompatible elastomeric coatings using 3D printer. However, there are some concerns need to be addressed.

  1. More information about the previous research is needed in the introduction part. What is the originality of your research?
  2. In result part, you need to add the characterization of the material.
  3. In figure 3, could you provide a control image, to compare the seeded cells.
  4. In figure 5(b), there is a huge decrease of viability in 3:37-40 UV 22h. Any explanation about this result?
  5. More mechanical explanation is needed in the discussion part.

Author Response

Answers to Reviewer 2

This paper provides a novel way to fabricate biocompatible elastomeric coatings using 3D printer. However, there are some concerns need to be addressed.

Comment R2.1 - More information about the previous research is needed in the introduction part. What is the originality of your research?

Answer to R2.1 The Introduction part of the manuscript was amended according to the recommendation of Reviewer.A description of various attempts addressed to mitigate the cytotoxicity of polymers was added.

Comment R2.2 - In result part, you need to add the characterization of the material.

Answer to R2.2 Mechanical properties of the material was added to the Materials and Methods part - a Table 1 was inserted.

Comment R2.3 In figure 3, could you provide a control image, to compare the seeded cells.

Answer to R2.3 We did not included the image of control cells grown on TCPS cells (Figure R1) as the SEM images were made only for qualitative analysis. In the manuscript we have provided representative images of cells grown on tested surfaces. All cells in the images are spindle shaped. There are only few round cells. This shows that the surfaces are biocompatible with cells and they are tend to multiply. These details were added into the caption of the text.

Comment R2.4 - In figure 5(b), there is a huge decrease of viability in 3:37-40 UV 22h. Any explanation about this result?

Answer to R2.4 Figure 5 shows distribution of viable, viable apoptotic, non-viable apoptotic and necrotic cells in the cell population in percent. Overall cell viability is shown in Figure 4. A higher decrease of cell viability was detected after 4 h sample processing. In our opinion, shorter soaking time might result in residual monomers that have effect of cell viability. On the other hand, a huge decrease of cell viability was registered on day 7. This effect might be associated with release of toxic compounds from polymer due to its biodegradation or erosion at a critical time point reported by Davison et al. in Tissue engineering 2014, p. 177-215. Interestingly, cell viability was reestablished on day 8. We hypothesize, our samples‘ processing procedure might overcome this critical time point.

Comment R2.5 - More mechanical explanation is needed in the discussion part.

Answer to R2.5 The Discussion part of the manuscript was amended according to the recommendation of Reviewer. Description of the biocompatibility requirements, effects of polymers and commercial resins on cell fate as well as impact of combinative treatment on structured surfaces were included in this part.

See R1 figure within the attached PDF file.

Figure R1: SEM analysis of myogenic stem cells grown on TCPS.

Reviewer 3 Report

The manuscript briefly presents cost effective ways to enhance elastomer biocompatibility which include prolonged immersion in polar solvents and additional UV exposure.

This protocol might be particularly interesting for improving the biocompatibility of 3D-printed polymer parts/coatings was introduced and experimentally validated.

The effects of this protocol on the porosity and mechanical properties of the printed materials would have been a plus.

I recommend this manuscript for publication in "Coatings", in the present form.

Author Response

Answers to Reviewer 3

The manuscript briefly presents cost effective ways to enhance elastomer biocompatibility which include prolonged immersion in polar solvents and additional UV exposure. This protocol might be particularly interesting for improving the biocompatibility of 3D-printed polymer parts/coatings was introduced and experimentally validated. The effects of this protocol on the porosity and mechanical properties of the printed materials would have been a plus.

I recommend this manuscript for publication in ”Coatings”, in the present form.

Answer to R3.0 We thank the Reviewer 3 for its dedicated attention towards our manuscript, we are very happy to see it was well understood and the research findings seems to be useful. The effects of this protocol on the porosity and its mechanical properties of the 3D printed materials will be our future study.

Reviewer 4 Report

I commend the authors for their research on commercially available 3D printer materials fabricated from SLA based 3D printers. Introduction was brief. All materials and methods used were enough described. Tables, figures and graphs were well elaborated. Conclusions drawn from the results were logical. I would suggest publishing the article in its present form.

Author Response

Answers to Reviewer 4

I commend the authors for their research on commercially available 3D printer materials fabricated from SLA based 3D printers. Introduction was brief. All materials and methods used were enough described. Tables, figures and graphs were well elaborated. Conclusions drawn from the results were logical. I would suggest publishing the article in its present form.

Answer to R4.0 We thank the Reviewer 4 for its dedicated attention towards our manuscript, we are very happy to see it was well understood and the research findings seems to be useful.

Round 2

Reviewer 2 Report

The current version of the article is quite well written. All concerns have been well addressed.